# The Determinants of Infant Mortality in Brazil, 2010–2020: A Scoping Review

**DOI:** 10.3390/ijerph18126464

**Published:** 2021-06-15

**Authors:** Alexandre Bugelli, Roxane Borgès Da Silva, Ladislau Dowbor, Claude Sicotte

**Affiliations:** 1École de Santé Publique de l’Université de Montréal (ESPUM), 7101, Park Avenue, 3rd Floor, Montreal, QC H3N 1X9, Canada; roxane.borges.da.silva@umontreal.ca (R.B.D.S.); claude.sicotte@umontreal.ca (C.S.); 2Centre de Recherche en Santé Publique (CReSP), École de Santé Publique, Université de Montréal, 7101, Park Avenue, 3rd Floor, Montreal, QC H3N 1X9, Canada; 3Science without Borders Program, CAPES Foundation (Coordenação de Aperfeiçoamento de Pessoal de Nível Superior), Ministry of Education of Brazil, Brasilia DF 700040-020, Brazil; 4School of Economics and Business Administration Graduate Program, Pontifícia Universidade Católica de São Paulo (PUCSP), Rua Monte Alegre, 984, Perdizes, São Paulo CEP 05014-901, Brazil; ladislau@dowbor.org

**Keywords:** infant mortality, health capabilities, public policies, social determinants of health, conditional cash transfer program, *Bolsa Familia* Program, primary care, Family Health Strategy

## Abstract

Despite the implementation of social and health policies that positively affected the health of the populations in Brazil, since 2009 the country has experienced a slower decline of infant mortality. After an economic and political crisis, Brazil witnessed increases in infant mortality that raised questions about what are the determinants of infant mortality after the implementation of such policies. We conducted a scoping review to identify and summarize those determinants with searches in three databases: LILACS, MEDLINE, and SCIELO. We included studies published between 2010 and 2020. We selected 23 papers: 83% associated infant mortality with public policies; 78% related infant mortality with the use of the health system and socioeconomic and living conditions; and 27% related to individual characteristics to infant mortality. Inequalities in the access to healthcare seem to have important implications in reducing infant mortality. Socioeconomic conditions and health-related factors such as income, education, fertility, housing, and the *Bolsa Família*. Program coverage was pointed out as the main determinants of infant mortality. Likewise, recent changes in infant mortality in Brazil are likely related to these factors. We also identified a gap in terms of studies on a possible association between employment and infant mortality.

## 1. Introduction

The infant mortality rate is a reliable indicator of population health and effectiveness of health systems that is also capable of estimating the extent of social and health inequalities between populations [1,2,3]. Despite the implementation of a set of social and health policies that positively affected the health of population [4], since 2009, Brazil has been experiencing a slower decline in infant mortality [5] that has remained at high levels and presents significant regional disparities. In 2016, the country recorded an increase in the mortality of children under one and under five years old, which disrupted a 25-year downward trend [5,6].

In 1994, the Brazilian Ministry of Health created the Family Health Program (now operating under the name of Family Health Strategy—FHS), which was a decentralized program based on primary healthcare that sent healthcare professionals into communities [4]. Created in 2003, the *Bolsa Família* Program (BFP) provides monthly cash transfers to poor families on the condition that they meet the program’s health and educational conditionalities. The health conditionalities established that parents were required to ensure that children younger than seven years of age had to comply with a routine of check-ups and growth monitoring and a childhood vaccination program. Pregnant women and nursing mothers were expected to be engaged in care and nutritional education programs at their local healthcare provider [7].

Since their implementation both programs, health indicators have improved such as life expectancy, maternal and infant mortality, and mortality due to transmissible diseases [8].

After an economic downturn that evolved into a troubled period of political crisis, in 2016 many regions reported an increase in the infant mortality rates [5,9,10].

Thus, these perturbations in infant mortality rates raise questions about the determinants of infant mortality in Brazil under the influence of such social and health policies. According to the World Health Organization (WHO), maternal and child health are closely related to social determinants of health that go beyond the impacts of adequate healthcare. Thus, infant health is also influenced by non-healthcare policies targeting socioeconomic and living conditions, which are as important as health policies for infant survival.

We conducted a scoping review to identify and summarize the determinants of infant mortality in Brazil under the influence of the FHS and BFP, with a view to raising hypotheses for the recent changes in the infant mortality rates and identifying gaps in terms of research concerning the determinants that may impact infant mortality in Brazil.

## 2. Material and Methods

### 2.1. Scoping Review Framework

The methodological approach employed was the framework proposed by Arksey and O’Malley [11], which breaks down the scoping review into 5 steps: Stage 1. Identifying the research purpose; Stage 2. Identifying relevant studies; Stage 3. Study selection; Stage 4. Charting the data, and Stage 5. Collating, summarizing, and reporting the results. In line with this approach, the steps above allow for reviewing the existing literature and examining the extent, scope, and nature of research activities on a given subject, in addition to identifying gaps, summarizing, and disseminating research results.

### 2.2. Identifying the Research Question

This review aimed to answer the following question: what are the determinants of infant mortality in Brazil under the influence of such social and health policies? We had two objectives:To raise hypotheses for the recent changes in the infant mortality rates in Brazil.Identifying gaps in terms of research concerning the determinants that may impact infant mortality in Brazil.

### 2.3. Identifying the Relevant Studies

#### 2.3.1. Inclusion and Exclusion Criteria

We included indexed quantitative studies on infant mortality by preventable causes according to the Brazilian List of Causes of Avoidable Deaths by Interventions of the Unified Health System (*Sistema Único de Saúde*—SUS), as follows: (a) avoidable by immunoprevention actions, (b) avoidable by providing adequate care to women during pregnancy and childbirth and to the fetus and newborn, (c) avoidable by appropriate diagnostic and treatment actions, and (d) avoidable by appropriate healthcare promotion actions linked to appropriate healthcare actions [12,13].

We sought studies on the determinants of the following indicators: neonatal mortality rate (NMR; between 0 and 27 days of life), early neonatal mortality rate (ENMR; between 0 and 6 days of life), late neonatal mortality rate (LNMR; between 7 and 27 days of life), under-1 infant mortality rate (IMR; between 0 and 1 year of age), and under-5 child mortality rate (U5MR; between 0 and 5 years of age).

The research interval included studies published between 2010 and 2020, with observation intervals that ended after 2004, when the current social and health policies were already in place. The searches were conducted from 4th January to 5th February 2020 and the selection was restricted to studies written in French, English, Portuguese, and Spanish. We established the country, the federal states, the five macro-regions, and the municipalities in their entirety or a representative proportion of the national territory and/or population as study units. Although it is unusual to have regional and methodological restrictions as exclusion criteria in scoping reviews, Armstrong et al. [14] suggest that regional and population limitations are valid tools to avoid selecting studies of low relevance.

We excluded the gray literature, editorial articles, letters from editors, correction letters, articles without a clear methodological approach, methodological analysis, opinion articles, quality assessment articles, data accuracy articles, and information systems analysis.

#### 2.3.2. Search Criteria

We searched for articles indexed in three databases: MEDLINE (U.S. National Library of Medicine), LILACS (Latin American and Caribbean Literature in Health Sciences), and SciELO (Scientific Electronic Library Online). Table 1 presents descriptors and keywords used in the research according to each database. The research equations were analyzed and reviewed by an expert library scientist.

## 3. Study Selection

In the first stage, duplicates were excluded. In the second stage, references were excluded based on title and abstracts with no relevance to the research objectives. In the third stage, a critical reading of the eligible articles was carried out, respecting the following inclusion criteria: study unit, population, relevance of the study to our research purposes, and methodology.

We used the Mixed Methods Appraisal Tool (MMAT) grid to evaluate the quality of the selected studies (Appendix B). The MMAT was designed as a critical checklist to provide a quality appraisal tool for quantitative, qualitative, and mixed methods studies.

This grid is quite complete and at the same time easily adaptable for the inclusion of new fields and information such as indicators of child mortality. The use of this tool is suitable for many types of quantitative studies in health, as it is not only focused on randomized or case–control studies but allows the use in quantitative research also based on literature reviews and surveys [15,16]. The use of the MMAT allowed us to assess eligible articles in order to identify and select those capable to provide evidence to answer our research questions. We established a minimum score of 80% in terms of methodological quality for an article to be included in this scoping review. The reading grid assessed studies characteristics regarding if there is an explicit methodological approach, clear objectives, and research purposes, if there is a clear explanation of variables, if data is likely to answer the research questions, and if they are complete.

### 3.1. Selected Studies

As seen in Figure 1, a total of 4236 titles were identified in the three databases. A total of 1484 duplicates were eliminated, and 2453 publications were excluded because of lack of relevance by title. A total of 299 articles were retained and 234 were eliminated because of lack of relevance by the abstract. Finally, after reading 65 eligible papers, 23 studies fully met the selection criteria.

### 3.2. Charting the Data, Summarizing and Reporting the Findings

To extract data from articles, to organize and to provide logical sense to our findings in respect to the Brazilian context, we used the Conceptual Model of Health Capability (CMHC) developed by Ruger [17] (Figure 2). This theoretical model based on the concept of capabilities of the Nobel Prize in Economics Amartya Sen [18,19] served as a guide for the interpretation of results in line with a current view of the social determinants of health.

According to Nussbaum, under the perspective of the capabilities approach, rights are understood in a positive way for which they require affirmative government support for their creation and preservation [20]. After enacting the 1988 Constitution, Brazil has established health as a basic human right and an obligation of the State. In this sense, social programs such as BFP are designed to motivate people to seek health and educational services through monetary incentives in exchange for families observing the program’s conditionalities [7].

In this same line, the central idea of the CMHC is that individuals seek both health and the ability to seek health. This conceptual framework considers the individual’s sense of health and functional capacity to attain health capability as the result of the interaction of four dimensions. An external dimension that refers to the macro, social, political, and economic environment, a second external dimension related to the effective use of the health services system, an intermediate dimension referring to the social and life contexts, and an internal dimension corresponding to the individual’s biologic and genetic predisposition to health/disease. In this framework, there is a fine line between State paternalism and self-agency as drivers to an individual, or a population, for pursuing and maintaining health as social and economic values. The concept of health capabilities has become increasingly important as an approach for assessing health. Further, infant mortality is considered as an appropriate indicator of population health attainment (health functioning), while the social determinants of health, such as education, housing, employment and economic inequalities are social and environmental conversion factors (capabilities) [21].

As a useful resource applied to other studies of the determinants of health that used an adapted framework on the determinants of health [22], we introduced some changes in the original CMHC framework, in order to identify the factors that are likely to effect on infant mortality in Brazil. In relation to the intermediate dimension, we used the concept of living conditions in a broader sense. It means that, in addition to housing, sanitation, safe water supply and income, we also consider poverty, socioeconomic inequalities, nutrition status, teenage pregnancy, late pregnancy, unemployment, fertility, culture, educational, and religious attainment as life and social contexts in the intermediate dimension. Although the CMHC was conceived aiming at the conceptualization and operationalization of health interventions at the individual level, the proposed version aims at identifying the determinants of infant mortality after the implementation of the FHS and BFP programs at the population level.

## 4. Results

### 4.1. The Determinants of Infant Mortality in Brazil

We begin our analysis by reporting all papers included (Table 2) and summarizing the findings according to the CMHC dimensions (Table 3 and Table 4), and then report the results.

Table 3 displays the frequency, as well as the proportion of articles selected according to the type of determinant, as defined by the four dimensions of the CMHC.

A majority of papers associated infant mortality with the effects of public policies of the external dimension 1 (83; *n* = 19). Most of studies focused on policies not directly related to health (65%; *n* = 15), while 57% of them associated infant mortality with health-related policies (*n* = 13). In the external dimension 2, the reorganization of health services was the most important factor impacting on infant mortality (78%; *n* = 18), with emphasis on the access and effective use of healthcare (61%; *n* = 14) and the use of primary healthcare (30%; *n* = 7). Access and quality of services were also the object of study in the selected articles (22%; *n* = 5). Studies reporting the effects of living conditions of the intermediary dimension on infant mortality rates, represent 78% of the total selected articles (*n* = 18), mostly related to income (35%; *n* = 8), housing (30%; *n* = 7), and education (30%; *n* = 7). Among the selected studies, few focused on individual factors and their effects on infant mortality rates (27%; *n* = 6).

Table 4 summarizes the results, indicating the main findings related to factors associated with infant mortality and their respective dimensions according to the CMHC.

### 4.2. The Four Dimensions of CMHC

In this subsection we reported the main findings according to the dimensions of the CMHC, the external dimensions 1 and 2, and the intermediate and the internal dimensions.

#### 4.2.1. External Dimension 1—Macro-Environment/Outcome of Public Policies

##### Healthcare Policies and Actions

The creation of the SUS and the implementation of the FHS were important factors to reduce infant mortality [29,31,34,35]. The decentralization of resources and autonomy in decision-making at the municipality level and greater availability of primary care physicians were also important for reductions in IMR [37,42]. On the other hand, decentralization and autonomy led to chronic underfunding, revealing the incapacity of small municipalities to provide access to adequate health by recruiting, retaining doctors, and hiring medium- and high-complexity procedures from the private health sector, leading to gaps in the comprehensiveness of services and increases in IMR [24]. Disparities in the degree of implementation of the SUS among regions and its effects on IMR [33] and U5MR [31], in addition to the rise of private health insurance coverage as a factor contributing for reducing IMR [37] raise questions about the lack of comprehensiveness and quality of public healthcare.

##### Policies Not Directly Related to Health

Although macroeconomic policies, such as a currency stabilization (*Plano Real*) and minimum wage policies, had led to reductions of PMR [32], socioeconomic inequalities remain [30,31,32] and were barriers to the access to health service system and its effectiveness to reduce infant mortality [4,23,33,37]. There was great variability in the evolution of socioeconomic inequalities on the decreasing trend in IMR and U5MR, suggesting that income inequalities impacted IMR and U5MR to a lesser extent over time [28].

##### The Brazilian Conditional Cash Transfer Bolsa Família Program

BFP was reported as a factor impacting infant mortality [7,27,29,32,35,37,42]. If, on the one hand, the FHS expanded the supply of primary healthcare, on the other, the PBF boosted the demand for the use of these services [7,29,35]. The effectiveness of FHS depended on the expansion of the BFP to reduce PNMR and the increased usage of prenatal care services was greater in the Northeast than in the other regions, although findings also raised the possibility that the programs may have been implemented in places where the decreases in PNMR were already underway [7,29]. The results also pointed out BFP as responsible for important increases on income and nutrition and the reduction of U5MR related to diarrhea [35].

#### 4.2.2. External Dimension 2—Access and Effective Use of the Health Service System

##### Coverage, Reorganization and Access to Health Services

Results confirmed the association of increased coverage of primary healthcare provided by the FHS with reductions in IMR [37] and U5MR [34], especially by preventable causes such as infectious diarrheic diseases [4]. Findings also demonstrated that higher coverage and improvements in primary healthcare were associated with reductions in infant mortality [7,27,29,35,42], contributing to reduce health inequalities. It was also reported that IMR is affected by the fact that health service facilities were concentrated in capitals, urban and central areas, which led to a significant disparity in providing health services to the rural, peripherical and poorest areas [23,33]. Disparities among macro-regions were related to the distribution of public health services and NMR [36]. Findings also reported that PNMR associated with complex cases depended strongly on referral structures [29], and that declines in IMR was also linked to increases of private health insurance coverage [37]. Although results reported increasing coverage of programs allowing access to primary healthcare, inequalities still remain, redirecting patients to emergency services and unnecessary hospitalizations, leading to mismanagement of cases relating to IMR and U5MR [31]. This also led to high rates of unnecessary caesarean sections and NMR [40].

##### Quality of Services

Results reported that approximately 70% of IMR are determined in the neonatal period, linking these deaths to the quality of prenatal care that is affected by gaps in the access to healthcare facilities [37]. The evolution of childbirth in Brazil was likely related to problems of hospital quality and antenatal care, and increases in the coverage of antenatal obstetric care and prenatal consultations were observed simultaneously with increases of preventable infant deaths (PMR and IMR) linked to adequate assistance and care during childbirth [40]. The literature also revealed that the prevention of infant mortality depended strongly on the preventive suggestions provided to families by the FHS teams and on the incorporation of these guidelines in seeking care at the appropriate time [42].

#### 4.2.3. Intermediate Dimension—Living Conditions, Social, and Life Contexts

##### Income, Poverty and Nutritional Status

The relation between per capita income and infant mortality is abundant in the literature [23,26,28,29,32,34,35,37]. The distances between housing and maternal, and child healthcare were inversely related to household per capita income, revealing low income as a driver of access inequalities linked to IMR [23]. Results revealed that living conditions, social and life context may affect infant mortality due to diarrhea through nutritional status related to poverty [26,27,33,35]. An association was found between IMR and dependency ratio (the proportion of people between 0 and 14 years old and/or 60 years old or over in relation to the total number of people between 15 and 59 years old living in a micro-region) [26,33]. Findings suggested that the elderly were responsible for the children and/or were providers of the very few financial resources of the household, most resulting from social benefits, which could lead to malnutrition and diarrhea [26].

##### Housing

Papers associated U5MR with adequate basic sanitation [27] and access to clean water with PNMR [29] and U5MR [34]. Garbage collection and piped water were associated with IMR [37], while declines in IMR were positively associated with the limited access to basic sanitation services [26,33].

##### Educational Attainment and Fertility

The literature reported education as inversely related to PNMR [29,33] and IMR [35,39], while U5MR was as inversely related to maternal education [28,38]. The greater the access to healthcare among social groups of higher income, the higher the schooling level and access to public services (such as water, electricity, sewage, and garbage collection services). Higher schooling level also favored a better perception of health and knowledge about the different medical specialties available for the treatment of diseases [32]. Results also reported an association between decreasing fertility rates and decreasing infant deaths [29,34,35,40,41].

##### Living Conditions and Development Indexes

Socioeconomic and living conditions composite indexes such as the Human Development Index [42], Municipal Human Development Index (MHDI) [30], and Family Development Index (FDI) [33] were associated with reductions in infant mortality. In addition to income as an important factor for increasing nutritional status, income inequalities that played an important role as a driver of inequalities in the access to healthcare because of an uneven provision of maternal and childcare [23]. The access to employment, adequate sanitation, and access to clean water, as components of FDI were pointed out as factors influencing infant mortality as well [33]. Higher educational attainment was identified as an element favoring a better perception of the health and knowledge about the treatment of diseases [32].

#### 4.2.4. Internal Dimension—Individual Characteristics/Genetic and Biological Factors

##### Low Birth Weight, Maternal Age, and Congenital Malformations

Prematurity and low birth weight (LBW) were associated with increased risk of U5MR [13]. In the opposite direction, higher LBW and lower IMR were observed in more developed regions when compared to less developed regions of the country. In fact, an epidemiological paradox was reported involving LBW, maternal age, and schooling [39]. There was an “age effect” regarding maternal schooling. Paradoxically, maternal age equal to or over 35 years in more developed contexts was associated with LBW and with low mortality rates. In these contexts, many women in this age bracket had a higher educational level and access to qualified jobs, better income, and access to better healthcare [32]. On the other hand, in underprivileged populations, maternal age equal to or over 35 years increased the risk of LBW and early neonatal mortality, both associated with low education and biological factors. In short, maternal schooling could hide an effect of socioeconomic condition on maternal age and be associated with LBW [13]. Access to better healthcare, interrupted pregnancy and high rates of caesarean sections may also contribute to low-birth-weight paradox (LBWP) [32,39].

Congenital anomalies [27] and malformations were also reported in association with prematurity [33] and NMR [7] and was observed as an increasing cause of infant deaths in the states with lower infant mortality rates, approaching the infant mortality profile of high-income countries [27].

## 5. Discussion

The objective of this review was to identify and summarize the determinants of infant mortality in Brazil under the influence of social and health policies, aiming at explaining the recent changes in the infant mortality rate and identifying gaps in research concerning such determinants.

As a summary of the findings related to the four dimensions of the CMHC, many improvements emerged with the implementation of the SUS, such as decentralization of services and resources that created autonomy in decision-making among states and municipalities [37,42]. Simultaneously, even under the effect of the FHS and BFP that intervened in both the supply and demand side of primary healthcare [29], decentralization led to disparities in the provision of health services among regions. The difficulties for small municipalities to hire private health services to close gaps in the provision of public services [24], coupled with a rise of private health insurance coverage and the availability of physicians associated with reductions in IMR raise questions about shortcomings in the access and comprehensiveness of public health services [23,43].

The increasing coverage of FHS accounted for important reductions in IMR and U5MR [34,37], mainly due to preventable causes such as diarrheic diseases [4], but deficiencies in the organization of health services, mainly in the distribution of maternal, child, and obstetric care, remained as marked inequalities between urban and rural regions [23,33]. The quality of health services was also a matter of concern, since, paradoxically, increases in the coverage of antenatal obstetric care and prenatal consultations were associated with increases of preventable infant deaths (PMR and IMR) linked to adequate assistance and care during childbirth [40].

Socioeconomic conditions also represented important factors impacting infant mortality, through income, education, and employment [4,7,28]. Even though macroeconomic policies had led to reductions of PNMR, socioeconomic inequalities have remained as barriers to the effectiveness of the health system [30,31,32] and policies to reduce infant mortality [4,23,33,37]. With regard individual factors linked to the internal dimension of the CMHC, the results suggest that age and maternal education are factors subject to confusion when associated with low birth weight and infant mortality rates [39]. The results indicate that congenital malformations are more prevalent in states with lower infant mortality rates [27].

This review also identified a gap in terms of studies on a possible direct relationship between employment and infant mortality.

What emerges from the literature suggests that public policies such as the implementation of SUS, the FHS, and BFP have proved to be important infant mortality reducers; however, some limitations related to inequalities in the access to quality and comprehensive health services that seem to have important implications for reducing infant mortality rates. Socioeconomic and living condition factors such as income, educational attainment, fertility rate, and housing were pointed out as the main determinants of infant mortality as well. Likewise, recent changes in infant mortality in Brazil are likely related to changes in one or more of those determinants. This study also shed light on the limited capacity of social and health policies in promoting sustainable reductions in infant mortality in Brazil in the presence of socioeconomic inequalities. We also identified a gap in terms of studies on a possible direct association between employment and infant mortality.

Our results are in accordance with other systematic reviews. Ferrari [44] reported that life conditions according to socioeconomic indicators such as income, housing, basic sanitation, and accessibility to health were identified as determinants of post-neonatal deaths. This review also highlighted the existence socioeconomic inequalities in the North and Northeast regions impacting on infant mortality. Duarte [45] summarized research studies from 1998 to 2006, aiming at assessing how Brazilian literature analyzed the infant mortality trends and possible associations with changes in the organization and financing of SUS. The review concluded that the impact of assistance and health measures on child mortality is limited, causing a reduction to a level that tends not to be exceeded unless they affect existing social inequalities. Santos [46] identified the main risk factors for infant mortality and the causes of death between 1980 and 1983 as well as between 2005 and 2008. In the first period, the main risk factors for infant mortality were related to socioeconomic characteristics and, in the second period, to the newborn, to maternal and child healthcare, and socioeconomic characteristics. This study concluded that in previous decades the priority was to solve problems linked mainly to the physical and social environment where the child lived, while in the second period the challenge incorporated the need to provide equity in access to quality health services.

### 5.1. The Implementation of FHS and BFP and the CMHC

Public policies have improved the provision of primary healthcare, operating as the gateway to public health services, through monetary incentives in exchange for the accomplishment of health conditionalities [7,29,34,35]. However, socioeconomic inequalities imposed limitations on the access to quality, comprehensive, and adequate health services, limiting the opportunities for seeking health once they cancel the internal features of a person (capabilities). These capabilities are fundamental in determining the degree of freedom for seeking health, such as self-knowledge, skills, and competences [47]. In such contexts, income appears as a fundamental element for increasing nutritional status [26,27,35], and also as an antidote against those inequalities [23].

The LBWP summarizes one of the mechanisms through which socioeconomic factors, such as income, educational attainment, and employment reinforce health inequalities that may impact maternal and infant health [32,39]. In this sense, maternal education and its relation to maternal age should be used with caution when assessing the social determinants of health in specific socioeconomic contexts. On the other hand, studies also emphasized that the prevention of infant mortality depended strongly on the preventive suggestions provided to families by the FHS teams and on the incorporation of such guidelines in the search for care at the appropriate time [42]. This finding suggests that educational attainment is a fundamental capability that impact on other capabilities [20] for reducing infant mortality.

Regarding the recent changes in infant mortality in Brazil, it is probably due to changes in income, educational attainment, fertility rate, housing, the access and the quality of healthcare, coverage of BFP, among other multifactorial determinants of infant mortality in Brazil.

In this review, we also identified a gap in terms of studies about a possible direct effect of employment on infant mortality. More quantitative studies are needed to assess the impact of those determinants on infant deaths.

#### Strengths and Limits

First, to our knowledge, this is the first study to make use of the health capabilities approach in a scoping review on the determinants of infant mortality. Second, this is the first scoping review on the determinants of the different infant mortality indicators gathered in a single study. One aspect that must be considered when interpreting our findings refers to the fact that the original version of the CMHC was conceived aiming at conceptualization and operationalization of health interventions at the individual level. Finally, due to the broad scope of the topic addressed, this study limited the analysis to quantitative studies only. This can be a source of selection bias in relation to the absence of qualitative studies aimed at identifying the variables acting within the intermediate dimension of the CMHC.

## 6. Conclusions

The findings suggest that although the implementation of the SUS, the FHS, and BFP have proved to be important infant mortality reducers, inequalities in the access to quality, and comprehensive health services that seem to have important implications for reducing infant mortality rates. Socioeconomic conditions and health-related factors interacting in the four dimensions of the CMHC such as income, educational attainment, fertility rate, housing, access to healthcare, and the BFP coverage rate were pointed out as the main determinants of infant mortality. Likewise, recent changes in infant mortality in Brazil are likely related to changes in those factors. This study also shed light on the limited capacity of social and health policies in promoting sustainable reductions in infant mortality in Brazil, mainly in the presence of socioeconomic inequalities. We also identified a gap in terms of studies on a possible direct relationship between employment and infant mortality. More quantitative studies are needed to assess the impact of those determinants on infant deaths.

With regard to the support offered by the findings of the current work to decision-makers, aiming at interventions for reducing infant mortality rates in Brazil, we provide some recommendations.

The quality of and access to health services are bottlenecks identified in the current work as limiters for reducing child mortality. Interventions and investments in actions that could improve the quality of and access to health services, whether through new methodologies for training personnel or in the form of remuneration of services, could improve the performance of the health service system.

Additionally, the resumption of the *Mais Médicos* (More Doctors) Program, which imported to Brazil doctors specialized in primary health care from other Latin American countries could reduce the shortage of health professionals in the most remote regions of the country. Investments in technology and infrastructure that promote digital inclusion would bring enormous benefits for the individual’s health management.

Considering that one of the most ambitious objectives of the BFP is to interrupt the intergenerational cycle of poverty in Brazil, the investment in public policies aimed at improving the quality and performance of the educational system, emerges as an element capable of acting at the same time in various dimensions of the Brazilian context. In addition to allowing the most effective use of public services by the beneficiaries of social and health policies, higher levels of education also mean more qualified workers and higher wages.

The comprehensiveness of services is a more complex problem to be addressed, as it arises as a result of the decentralization of resources and services. Given the high number of existing municipalities in the country (5565), this factor would be subject to a political predisposition towards a broad administrative reform that includes the reduction of health regions, together with a reduction in the number of municipalities and a reorganization of services. Further, there still exist political developments of the recent economic and political crisis in Brazil that impose major difficulties to public health managers regarding possible reforms and new proposals aimed at greater social protection and improvement in the population health. Furthermore, the emergence of the COVID-19 pandemic poses greater challenges for emerging countries such as Brazil, that may hinder efforts towards reforms in the health system.

## Figures and Tables

**Figure 1 ijerph-18-06464-f001:**
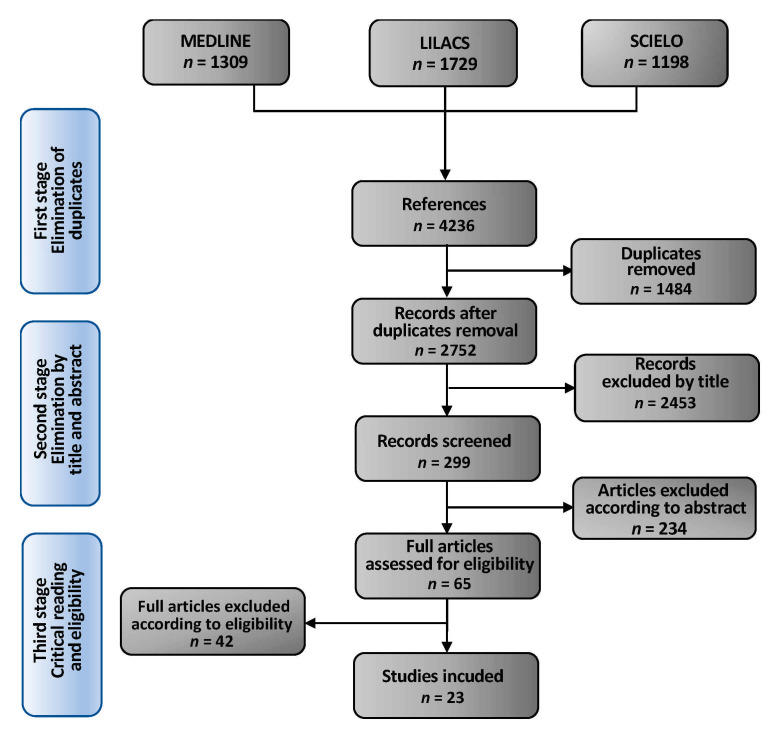
Study selection flow chart.

**Figure 2 ijerph-18-06464-f002:**
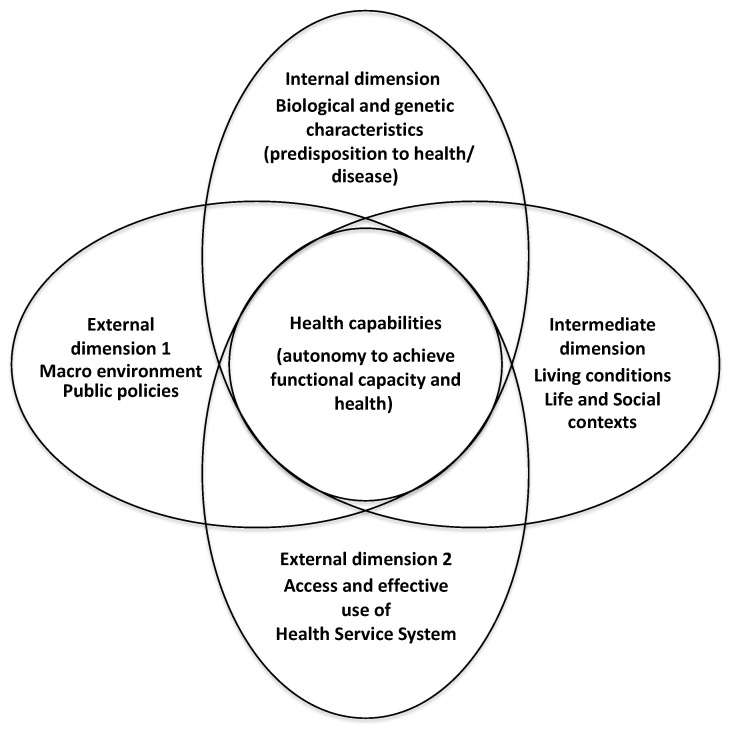
Adapted from Conceptual Model of Health Capability (CMHC–Ruger, 2010) [17].

**Table 1 ijerph-18-06464-t001:** Descriptors and keywords used according to database ^†^.

MEDLINE	MeSh ** descriptors: ((mortali* or death* or fatali*) adj3 (neonatal* or neo natal* or new born* or newborn* or infant* or child* or baby* or babies* or kid* or kids* or paediatric* or pediatric*)). ab, kf, kw, ti. Keywords: “death” “fatality”, “neonatal”, “neo natal”, “newborn”, “new born”, “child”, “baby”, “babies”, “kid”, “kids”, “paediatric” and “pediatric”
LILACS	Descriptors: Concept 1: mh:(mh:(((((mortality OR death OR “cause of death”) AND (child OR infant OR “infant, newborn”)) OR (“child mortality” OR “infant mortality” OR “perinatal mortality”)) AND (brazil))) AND (db:(“LILACS”)) AND (year_cluster: [2010 TO 2020]))” Keywords: w:((mortali* OR death* OR fatali*) AND (neonat* OR “neonat*” OR newborn* OR “new born*” OR “recém nascido*” OR infant* OR child* OR crianca AND (brasil* OR brazil*)) AND (db:(“LILACS”)) AND (year_cluster:[2010 TO 2020]) AND (db:(“LILACS”))
SCIELO	Keywords: (mortali* OR death* OR fatalit*) AND (neonatal* OR “neonatal*” OR “new born*” OR newborn* OR infant* OR child* OR crianca OR “recém nascido”) AND (brasil* OR brazil*) AND year_cluster:(“2013” OR “2014” OR “2019” OR “2017” OR “2011” OR “2018” OR “2010” OR “2016” OR “2012” OR “2015”)

1. ^†^ Database descriptors, keywords, and search equations are fully described in Appendix A; 2. ** MeSH refers to Medical Subject Headings database descriptors; 3. “ab” = abstract; 4. “kf” = keyword heading word; 5. “kw” = keyword heading; 6. “ti” = title; 7. mh = MeSH; 8. “db” = database; 9. “w” = keywords.

**Table 2 ijerph-18-06464-t002:** Studies included by period, methods, sample/study unit, infant mortality indicator and study objective.

Author	Period	Methods	Sample/Study Unit	Indicator	Objective
Almeida, W. [23]	2005–2007	Ecological study with geospatial analysis	Newborns and deceased infants in the 5564 municipalities	IMR	To analyze geographic access to childbirth in hospital in Brazil municipalities and IMR
Araújo, C. [24]	2010	Retrospective descriptive analysis	Deceased children of mothers living in 5526 municipalities	IMR	To evaluate the effect of municipal per capita spending on health on IMR
Boschi-Pinto, C. [25]	1990–2015	Ecological study	Populations and regions of 75 low and middle income countries with high burden of diarrhea and pneumonia, including Brazil	U5MR	To explore whether the adoption of national policies for the management of pneumonia and diarrhea is associated with the decline U5MR
Bühler, H. [26]	2010	Ecological study with geospatial analysis	Deceased children under-one year from mothers who lived in the 558 health micro-regions	IMR	To study environmental indicators for diarrhea in children under one year of age in Brazil and IMR
França, E. [27]	1990–2015	Ecological study and statistical analysis	Deceased children under-five and general population	U5MR	To analyze the leading causes of U5MR, using estimates from the Global Burden of Disease Study (GBD) 2015
Garcia, L. [28]	1993–2008	Cross-sectional study	Macro-regions, units of the Federation and nine metropolitan regions	IMR/U5MR	To study the temporal evolution and the extent of inequalities in infant and child mortality
Gomes, T. [4]	2000–2011	Ecological study with time series	Under-five deceased children in Brazil and macro-regions	U5MR	To analyze the trends in childhood mortality in Brazil and regions study the correlation between acute diarrheal disease and acute respiratory infection and U5MR
Guanais, F. [29]	1998–2010	Panel data	Deceased children of families benefiting from the BFP and FHS living in 4853 municipalities	NMR/Post-neonatal mortality rate (PNMR)	To examine the combined effects of (FHS) and BFP on NMR and PNMR
Malta, D. [13]	2000–2013	Ecological study with time series	All under-five children deceased in Brazil and macro-regions	U5MR	To analyze the trend in U5MR according to the list of preventable causes of death
Martins, P. [30]	2000–2010	Ecological study	Children under-one deceased from mothers living in the states and macro-regions	IMR	To analyze the convergence between the decrease in IMR and the Municipal Human Development Index
Mendes, P. [31]	2000–2010	Ecological study with time series	Under-one and under-five deceased children in the 5 macro-regions	IIMR/U5MR	To analyze the temporal trends of indicators of IMR and U5MR related to hospital morbidity due to diarrheal diseases
Oliveira, G. [32]	2006–2010	Ecological study with geospatial analysis	Newborns deceased in the 26 states and the Federal District, Brasília	NMR	To analyze the spatial distribution of neonatal mortality and its correlation with biological, socioeconomic, maternal and child factors
Ramalho, W. [33]	2006–2008	Ecological study	Children deceased between 27th and the 364th day of life in the 5227	NMR/ENMR/LNMR/PNMR	To describe the inequalities in infant mortality according to socio-economic indicators between geographic areas and municipalities in Brazil
Rasella, D. [34]	2000–2005	Panel data	Children under-five who died in Brazilian municipalities	NMR/IMR/PNMR/U5MR	To assess the effects of the FHS on the U5MR du to diarrhea diarrheal diseases and lower respiratory tract infections
Rasella, D. [35]	2004–2009	Panel data	Children under-five who died in 2853 municipalities.	U5MR	To assess the effect of BFP on deaths of children under-five, associated with to poverty, diarrhea, lower respiratory tract infections and malnutrition
Rodrigues, N. [36]	1997–2000/ 2001–2004/ 2005–2008/ 2009–2012	Ecological study with geospatial analysis	Deceased children from mothers living in the 5 macro-regions	ENMR/LNMR	To assess the spatial and temporal trends of maternaland neonatal mortality.
Russo, L. [37]	2005–2012	Panel data	Deceased children under-one year in 5563 municipalities	IMR	To study the effect of primary care physicians on IMR
Schuck-Paim, C. [38]	1980–2010	Retrospective descriptive analysis	Children under-five who died frompneumonia from mothers living in the 5570 municipalities	U5MR	To assess the effect of ten-valent pneumococcal conjugate vaccine (PCV10) on under-five mortality from pneumonia
Shei, A. [7]	1998–2008	Times series study	Infant deaths in the all municipalities in the country.	NMR/PNMR/IMR	To examine whether the implementation and expansion of the BFP Program, was associated with infant mortality
Silva, A.A. [39]	1995–2007	Correlational descriptive study	Deceased children in the five Macro-regions and the 26 states and the Federal District, Brasília	IMR	To examine whether the low birth weight (LBW) paradox exists in Brazil
Silva, A.L.D. [40]	1999–2013	Ecological time series	The country’s population (women of childbearing age, children born alive and deceased in the national territory	NMR/IMR	To analyze childbirth assistance according to birth profile, characteristics of live births and preventable infant deaths
Verona, A. [41]	1996–2006	Correlational analysis	Under-one children deceased from mothers aged 15 to 49 who had at least one child in the five years preceding the survey	IMR	To examine the relation between IMR and religious involvement of mothers
Vieira-Meyer, A. [42]	2012	Ecological study	Under-one deceased children from mothers living in 3441 municipalities	IMR	To access how the coverage and quality of FHS and BFP are associated with IMR

IMR: under-1 infant mortality rate; U5MR: under-5 child mortality rate; NMR: neonatal mortality rate; PNMR: post-neonatal mortality rate; ENMR: early neonatal mortality rate; LNMR: late neonatal mortality rate.

**Table 3 ijerph-18-06464-t003:** Frequency and classification of determinants of infant mortality according to the CMHC.

	Outcome of Public Policies	Health Policies	Policies Not Directly Related to Health	Conditional Cash Transfer Program	Reorganization of Health Service System	Access and Effective Use of Healthcare	Access to Health Services	Quality of Healthcare	Primary Healthcare	Living Conditions	Income	Housing	Education	Nutritional Status and Poverty	Fertiliy Rate	Inividual Characteristics	Maternal Age	Low Birth Weight	Congenital Malformation
External Dimension 1	19	13	15	7															
External Dimension 2					18	14	5	5	7										
Intermediate Dimension										18	8	7	7	5	5				
Internal Dimension																6	1	4	2
Proportion	83%	57%	65%	30%	78%	61%	22%	22%	30%	78%	35%	30%	30%	22%	22%	27%	4%	17%	9%

**Table 4 ijerph-18-06464-t004:** The determinants of infant mortality in Brazil according to the four dimensions of the Conceptual Model of Health Capability (CMHC).

Main Author/Year	External Dimension 1 *	External Dimension 2 *	Intermediate Dimension *	Internal Dimension *
Almeida, W./2012 [23]	Socioeconomic conditions	Unequal access and quality of healthcare/organization of healthcare	Geographical distance and the difficult access to childbirth facility/socioeconomic and cultural factors	_
Araújo, C./2017 [24]	Per capita spending of municipality’s own resources on healthcare	Unequal access to healthcare	Living conditions	_
Boschi-Pinto, C./2017 [25]	National policy of management for treating pneumonia and diarrhea/the Millennium Development Goal SDG-4	_	_	_
Bühler, H./2014 [26]	Socio-environmental policies	_	Percentage of residents without garbage collection service and dependency ratio	_
França, E./2017 [27]	BFP/National Immunization Program (NIP)/FHS	Primary healthcare/reorganization of prenatal and neonatal care	Improvements in nutrition/basic sanitation	Prematurity/congenital anomalies
Garcia, L./2011 [28]	Regional socioeconomic inequalities/household per capita income	_	Maternal schooling/living conditions	_
Gomes, T./2016 [4]	Expanding coverage rate of FHS/improvements in socioeconomic conditions	Increase in the population covered by primary care	_	_
Guanais, F./2013 [29]	The expansion and interaction between FHS and BFP programs	Quality of hospital birth care	Improvements in daily living Conditions/ fertility rate	_
Malta, D./2019 [13]	SUS/healthcare promotion actions linked to healthcare actions	Adequate neonatal care/diagnostic/therapeutic actions and care during childbirth	_	Short-term pregnancy/low birth weight (LBW)
Martins, P./2018 [30]	Regional socioeconomic disparities	_	Living conditions expressed by Municipal Human Development Index (MHDI)	_
Mendes, P./2013 [31]	Healthcare policies and socioeconomic inequalities	Limited access to healthcare	Socioeconomic and cultural disparities	_
Oliveira, G./2013 [32]	Macroeconomic policies/BFP/socioeconomic and regional inequalities	Inequalities in accessing maternal/prenatal/birth care and caesarean sections	Living conditions/maternaleducation/Low Birth Weight Paradox (LBWP)	Maternal age/teenage pregnancy/LBW
Ramalho, W./2013 [33]	Socioeconomic conditions measured by Family Development Index (FDI)	Coverage of healthcare/healthcare information system	FDI/family vulnerability/social mobilization	Congenital malformation
Rasella, D./2010 [34]	FHS coverage rate	Reorganization of primary healthcare/early case diagnosis/antibiotic prescription	socioeconomic conditions	_
Rasella, D./2013 [35]	BFP and FHS coverage rate	Increased primary care through BFP	Extreme poverty/undernutrition	_
Rodrigues, N./2016 [36]	_	Unequal distribution of healthcare among the macro-regions	_	_
Russo, L./2019 [37]	Gross Domestic Product per capita/FHS	Availability of primary care physician/private health insurance coverage	Gross Domestic Product (GDP) per capita/piped water/electricity/garbage collection	_
Schuck-Paim, C./2019 [38]	National Immunization Program	Vaccination/improved education and healthcare	Improved nutrition and hygiene/maternal education	_
Shei, A./2013 [7]	Expanding coverage rate of the PBF	Improved access to healthcare	Reduction of health inequalities	Congenital malformation
Silva, A.A./2010 [39]	_	Healthcare during pregnancy/early medical interventions	Socioeconomic conditions/maternal education	LBWP
Silva, A.L.D./2016 [40]	_	Hospital quality/increased use of health private sector	_	LBW
Verona, A./2010 [41]	_	_	Maternal religious involvement, parity and region	_
Vieira-Meyer, A./2019 [42]	FHS and PBF coverage rate	Quality and effectiveness of FHS	Socioeconomic conditions/Human Development Index/family attitude towards health	_

(*) As defined in the CMHC, external dimension 1 refers to the outcome of the macro-environment; external dimension 2 is related to the access and effective use of health system; the intermediate dimension is linked to social and life context, while the internal dimension refers to biological and genetic individual characteristics.

## Data Availability

The data presented in this study are openly available in https://udemontreal-my.sharepoint.com/:u:/g/personal/roxane_borges_da_silva_umontreal_ca/EaSSLZmDPt5KvE1SL2eUJPwB2c7IFSE8fBi72oSn_M33Sw?e=LxyibO (accessed on 12 June 2021).

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
