# Peer review of "The Determinants of Infant Mortality in Brazil, 2010–2020: A Scoping Review"

_ijerph, 2021, doi:10.3390/ijerph18126464_

Round 1

Reviewer 1 Report

Review Report:

Title: The determinants of infant mortality in Brazil, 2010-2020: a scoping review

There is no doubt it is really interesting to dig in what influence on infant mortality, however although the idea of combining different database and making a meta-analysis is good, it is difficult for the reader to follow the paper. In my opinion the authors must work in the way they present the method (here is the key) and the results, It is difficult to see how they reach a conclusion looking at the screenshots of the end of the paper

Reviewer 2 Report

The determinants of infant mortality in Brazil 2010-2020: a scoping review. The manuscript contained a good amount of information reviewed regarding infant mortality in Brazil. It shows obvious effort by the authors. I found a few spots with missing or misplaced articles (in/on), and a few extra lengthy sentences as such that I would recommend a person more familiar with English grammar to review the paper for these types of corrections. Additionally, I had hoped for some type of solid opinion or idea in the conclusion. I wonder what type of policies, be it in health or not, would the authors think are needed to address these issues? And how likely would positive outcomes result in their opinion? Unfortunately we as readers were left with a simple statement that more studies are needed. It’s unfortunate because I believe the authors could discern more from the data than they have presented. However, overall I found little else in need of correction and those specifics that do are listed below.

Page 2, line 51-53: This is an example of English grammar issues, “in” should be “at”

Page 2, line 88: What is SUS? There is no place with it spelled out prior to using abbreviations. This happens a few other times, thus I’d like to suggest the authors add an appendix with the many used abbreviations and what they stand for so that readers can quickly find this information.

Page 4, Fig 1: The figure description needs more information to clearly define what it represents.

Page 5, Fig 2: Same as previous comment. What does this figure show the reader?

Page 5, line 144-149: This is entirely one sentence. It is so long and includes so much information that I’d forgotten how the start of the sentence had begun. Please break up sentences >50 words, it becomes cumbersome. This type of structure occurs elsewhere in the text as well.

Page 5, line 162: The word “income” is listed 2x as considerations.

Page 7, Table 2: I first thought this was a figure but then tried to read it as a table. The issue is that the column headings are so long it gets confusing. Perhaps slant the text or widen the table and use wrapped text. It would also be helpful to put one or two vertical lines to help guide the readers eyes from top to bottom.

Table 3: The title needs more information as mentioned for the figures. Also, the table itself might work best if beginning a new page in landscape format to give more space for the information. And also add a line separation between each study so that the objectives don’t blend together.

Table 4: I completely missed this table as it is set in the same manner as the previous and has no distinct title information. It needs to stand apart more clearly.

Page 14, line 272-277 and Page 15, line 333-339: Examples of overly long sentences that need to be broken up so that information isn’t missed.

Page 14, paragraph 4.2.4: Why was the age 35 chosen? It’s not clear if this is Brasil specific or worldwide and for what reason.

Page 15, line 346-351: An overly long sentence in which the second part beginning after ref #42 should stand alone. This seems to be rather an important statement not to be left hidden at the end of such a long sentence.

Round 2

Reviewer 1 Report

The paper has been highly improved